# Positioning and Web Traffic of Colombian Banking Establishments

**Joan Sebastián Rojas Rincón** [1], **Andrés Ricardo Riveros Tarazona** [1], **Andrés Mauricio Mejía Martínez** [1] and **Julio César Acosta-Prado** [2,*]

1. School of Administrative, Accounting, Economic and Business Sciences, Universidad Nacional Abierta y a Distancia—UNAD, Bogotá 111511, Colombia
2. School of Business Science, Universidad del Pacífico, Lima 15072, Peru
* Correspondence: jc.acostap@up.edu.pe; Tel.: +51-1-219-0100 (ext. 2660)

**Abstract:** The use of digital technologies has become one factor that significantly impacts business results in the financial industry. This study seeks to characterize the positioning and web traffic of Colombian banking establishments through analysis of the classification of their website, taking as reference the metrics related to web traffic and the attractiveness of the content and relevance for users as the bounce rate. The study presents a quantitative approach, non-experimental design, and descriptive scope. With a sample of 28 banking establishments, it is intended to contribute to the body of literature on bank marketing based on a systematic analysis of indicators. The findings of the study made it possible to elucidate that a good part of the websites of the banking establishments is well positioned, in addition to presenting low bounce rates. It is also possible to show that a significant portion of this traffic comes from individuals between 18 and 34 years of age and of the female gender. Likewise, traffic to the website is derived to a greater extent from direct access to the establishment's portal or search engines.

**Keywords:** positioning; web traffic; banking establishments; digital marketing; banking marketing; digital channels

## 1. Introduction

Digital technologies have become a relevant competitive factor for organizations, given the intensive and widespread use of the Internet, mobile devices, social networks, and other features to which consumers have access today. Evidently, the financial industry has not been oblivious to this phenomenon, since there is an increasingly intense use of "mobile platforms and applications such as PayPal and Chime, online payment systems, digital currencies, robotic assistants and loans between individuals" [1] (p. 578). The development of a digital strategy has become an imperative for the growth and sustainability of credit establishments and other financial institutions, which induces that, in their budget plans, a more significant amount of resources is allocated to platforms, applications, tools, and channels that facilitate communication, interaction, and service delivery. According to Singh et al. [2], banks today look for unconventional ways to deliver and differentiate their various services. However, the incorporation or development of new technologies implies the availability of resources that have a cost; therefore, it is imperative to evaluate the effect that access to them can have in terms of competitiveness at the industry level and financial performance.

One of the main fields where new technologies are applied is in marketing, considering that this is a key function in the financial industry, in the sense that it is required to establish an image positioned around trust and service quality, to captivate the consumer. According to Leong et al. [3], with access to advanced technologies, it is possible to use text and images to explain to customers that continuous improvement in the company's innovation aims to ensure that their financial assets are safe in the organization. Financial innovation

can be defined as "the development, introduction, and management of an innovation (product, a process, a business model or a technology) to assist or facilitate one or more of the four main functions of the financial system" [1] (p. 584). These functions involve raising resources, transferring resources, managing and controlling risk, and providing information to the market for decision making. Considering these functions, many of the digital technologies introduced by organizations will be oriented to develop new transactional channels, information security systems, and resource-sharing schemes among users. The adoption of these new solutions, under the online banking model, "is more complex; as it generally involves a long-term relational exchange between the customer and the bank" (Lee [4], as cited in Oertzen and Odekerken-Schröder [5]) (p. 1395).

Understanding the business model in the financial sphere as the representation of how an institution creates, transfers, and delivers value to the user, it is possible to understand that the level of development of the digital strategy is directly related to the level of business competitiveness. However, as discussed previously, incorporating these digital technologies implies a severe challenge for financial institutions; by their nature, society has attributed an important fiduciary responsibility to credit institutions. Therefore, these institutions must ensure that these innovations not only improve the customer experience but also guarantee security in managing financial resources, in an environment of trust. Ennew and Waite [6] propose that, although any company has a responsibility to its consumers in terms of the quality, reliability, and security of the products it supplies, this responsibility is perhaps much more significant in the case of a financial service provider. Taking into account the role of trust in the value proposition offered by financial institutions, studies such as Kaabachi et al. [7] highlight the advantages of organizations that adopt an omnichannel model compared to those that develop their strategy exclusively on the Internet, since physical presence can improve the credibility of the company. For this reason, when implementing a digital strategy, it is essential for the managers of financial institutions to evaluate the impact on the security of users and the integrity of transactions.

In the Colombian context, digital models have been consolidated for the financial industry; proof of this is the exponential growth of Fintech and the entry into digital credit establishments such as Lulo Bank and Nu Bank. In this regard, Gutiérrez and Hurtado [8] point out that by 2018 there was a 90% increase in the use of ICT within the banking system, and 56% of the customers of the entities attached to the Superintendency of Finance made use of the network to perform monetary transactions. Similarly, the Financial Superintendency of Colombia (as cited in Cajamarca [9]) indicates that 52.8% of monetary and non-monetary transactions were made via cell phone, followed by the Internet with 20.1%; so, 72% of banking transactions are already made digitally. This vertiginous adoption of digital technologies was driven significantly by the COVID-19 pandemic, since even from the Ministry of Information and Communication Technologies came the express mandate to focus on telemedicine, telework, and mobile banking, the last of which to facilitate that financial services can reach Colombia more deeply [10,11]. The above reflects the importance that digital channels have today in the business strategy of Colombian banking; however, it is advisable for financial institutions to frequently evaluate the effect that the development of such channels has on customer satisfaction, loyalty, and business results. In addition, there are still relevant challenges regarding access to the financial system by Colombians because, as evidenced by Rodríguez-Raga and Riaño Rodríguez [12], the costs of using the financial system, in addition to being higher than in other countries, have been increasing, which could represent a barrier to access for families.

Based on the above, this study aims to characterize the positioning and web traffic of Colombian banking establishments. This is justified given the increase in the competitive intensity in the financial industry since, in addition to the introduction of digital financial institutions and the entry of international entities such as J.P. Morgan bank, the emerging Fintech models are exerting an increasing influence on the dynamics of the financial business. Thus, today, solutions offered by financial institutions are more familiar to citizens, among which we can mention "Bancolombia a la Mano, Daviplata and Movii, as

leading financial technologies that offer this service in Colombia" [11] (p. 2). However, incorporating these technologies as a transactional and communication channel with the Colombian financial user confronts credit establishments with new challenges related to the nature of digital technologies and financial products. Therefore, it is expected that this study will provide evidence of relevant differences in the positioning of those banking establishments that make significant efforts in technological innovation concerning those that do not.

In this sense, there is a need to characterize the activity of banking establishments in digital media. According to Waite and Rowley [13], banks may have multiple web presences, sometimes even more than one website. Therefore, one way to approach this problem is, on the one hand, the characterization of positioning and, on the other hand, the traffic to the website of financial establishments.

## 2. Literature Review

Due to the growing importance of digitalization and technological convergence within the financial industry, it is possible to find a compendium of academic literature related to technological applications in financial services, especially as support for marketing activities. For example, in the study developed by Lee and Lee [14], the concept of contactless service is addressed, i.e., without the need for a personal encounter, since they can be enabled by advanced technologies such as ATMs, electronic banking, or online purchase and payment systems. According to what is proposed in this paper, "these new business models create value by changing human-human relationships to human-technology relationships" [14] (p. 7). Although the exponential increase of technologies makes them increasingly important, the study concludes that some consumers still prefer direct interactions with others and that value cannot be created by excluding humans; for this reason, if effective digital contact is to be developed, it will be necessary to deliver the required value of customer-oriented service. Regarding this value associated with customer experience, Komulainen and Saraniemi [15] adopt an approach that divides it into four main categories such as process, outcome, time, and place, where practices based on operations (utilitarian value), emotional dimensions (hedonic value), or social dimensions (customer's social environment) can be included. In their work, the authors seek to understand how customer value can be enhanced to improve the performance of banks and other financial institutions, taking advantage of the benefits of mobile banking. The results are obtained from a qualitative case study, where they conclude that mobile banking creates utilitarian and hedonic value but not social value, because the latter does not seem to play an important role in the context of banking services, "which are perceived as private and personal in nature" [15] (p. 1098).

Among the technologies that bring banking establishments closer to the customer, either as a transactional channel or as a means of communication, we can mention Internet banking, mobile banking, social networks, customer relationship management systems (CRM), artificial intelligence, and the Internet of things, among others. According to Nejad [1], while in the early days companies focused on using digital banking to attract new customers, others concentrated on strengthening relationships with existing customers by using Internet banking as an additional service. In the work of Oertzen and Odekerken-Schröder [5], the attitudes and behaviors of users are analyzed after an e-postbox was adopted as a functional feature of online banking. The service offers some modules intended to improve the user experience, such as bank statements, security documents, personal messaging, personalized offers, and notifications, thus being an innovation that this German bank "introduced to save time, be sustainable, be secure and build closer relationships with customers" [5] (p. 1403). According to the results of their study, receiving electronic voice-to-voice messages (WOM) induces positive customer attitudes towards the email inbox, but sending subsequent WOM is given in a negative way. Nazaritehrani and Mashali [16] analyze the effect that the development of innovative service delivery channels can have on market share, i.e., focusing on the ability to access new customers.

For the development of this study, the authors propose a structural equation model that includes variables such as mobile banking, ATMs, Internet banking, telephone banking, and point-of-sale (POS) terminals. The endogenous variable is growing in market share. The results show that there is a relationship between new banking service platforms and market share, providing evidence that "the development of three banking channels, including Internet banking, telephone banking, and POS terminals, have a significant and positive impact on market share growth" [16] (p. 13). In congruence with the previously mentioned studies, it is possible to observe a growing concern in the academic literature to analyze the effects that technologies have on the business performance and financial results of banking establishments, considering that in many works a statistically significant relationship between reputation and financial performance has become evident, and nowadays, "the growing prominence of social networks and the Internet, where bad news spreads faster, has put the reputation of companies at greater risk" [17] (p. 1400).

This discussion is also grounded in a disciplinary field that has been growing significantly and is called financial services marketing, also known as bank marketing. In this field, considerable emphasis is placed on the characteristics of financial services. According to Ennew and Waite [6], these services are characterized, among other things, by the following:

Intangibility: this means that the services cannot be touched, as they lack a substantive physical form and therefore cannot be perceived through the senses.

Inseparability: this means that services are produced and consumed simultaneously, given their nature of being a process or an experience.

Expiration: In the sense that services can only be produced when the consumer wishes to buy them. Unlike products, surplus services cannot be produced to be stored and sold when demand increases.

Heterogeneity: This can be interpreted in two ways. First, services are not standardized and generally tailored to specific needs. On the other hand, the way the customer perceives the service may be different, which is not so much related to changing needs, but because the interaction between customer and provider "may be influenced by events outside the control of the service provider" [6] (p. 59).

The four characteristics mentioned above are transversal to most services; however, some attributes are particular to financial services. According to Ennew and Waite [6], one of these attributes is fiduciary responsibility, related to the management of funds and advice, which turns out to be more important in the consumption of financial services concerning other services. The explanation for this is that, for many consumers, financial services are difficult to understand. The other fact is that the raw material used for the elaboration of the financial product is precisely the funds of third parties; therefore, banking institutions have a responsibility to the person who requests a loan, as well as to the depositors who have made that loan possible. Another characteristic is contingent consumption, which means that many financial services do not produce direct consumption but create opportunities for the future, although in some cases such consumption may never materialize. In the case, for example, of a pension fund, capital is being created that is expected to be used in the future. On the other hand, whoever purchases a life insurance policy, in most cases, is not looking for the benefit to materialize since it will be effective in the event of the insured's death. Another characteristic has to do with the duration of consumption since most services are extended to the long term or have the possibility of doing so, generally, through the establishment of a contractual relationship with which links can be created; however, "for such a relationship to be beneficial and for cross-selling opportunities to work, the organization has to work on the relationship" [6] (p. 64). In that order of ideas, it is possible to find a clear connection between the marketing management carried out in financial institutions and the relationship marketing approach.

Among the theoretical approaches that allow an approach to the objectives of marketing management is the one that bases them on the purpose of establishing a quality and lasting relationship between the agents participating in an exchange process. According

to what Reinares [18] proposes, relationship marketing is a strategic process based on identifying, capturing, and maintaining relationships with consumers and other agents belonging to the company's interest groups. In this regard, Ennew and Waite [6] argue that, in the financial industry, institutions have traditionally not managed relationships well in a mass market context; however, recently, a set of environmental factors has contributed to increased concern about these relationships. This concern is important since the value of financial services is related to good experiences and credibility, since "its quality is best judged after purchase <qualities of experience> but, even after purchase, it may be difficult to assess whether the quality obtained is the result of a good product or favorable supplier, given the economic circumstances <qualities of credibility>" [6] (p. 153).

Grounded in relationship marketing theory, Morgan and Hunt [19] propose that shared values, communication, and opportunistic behavior influence trust. Although trust is a crucial aspect of the financial field, it is also important to consider other dimensions. One of the crucial aspects is commitment, understood as the lasting desire to maintain a valuable relationship. Satisfaction with the relationship also has an influence, which, according to Ennew and Waite [6], is a pleasant result related to a desired final state of consumption. In Minta [20], it is evident that these variables can be linked, so that trust and commitment can act as moderators between satisfaction and loyalty. In this way, trust can amplify satisfaction since it reduces uncertainty, establishes credibility, and guarantees the stability of exchanges. Commitment, for its part, can strengthen the relationship between satisfaction and loyalty since it reinforces the customer's attitude towards a business, increasing the intention to cooperate; hence, it is "likely that interaction between strongly committed partners increases the satisfaction-loyalty effect" [2] (p. 27).

Considering the importance of relationship marketing in the management of financial institutions, Figure 1 illustrates the relationship between digital channels and the performance of credit institutions. As discussed in the previous lines, marketing management aimed at building long-lasting relationships must be supported by attributes such as trust and commitment; therefore, it is expected that the development of digital channels operationalizes the management, articulate actions, means, and technologies that contribute to improving customer satisfaction with a value proposition that enjoys such attributes. For example, Abid et al. [21] point to the case of social networks, where some influencers with some expertise in specific areas can be identified who attract many followers and therefore can offer greater confidence in messages to a wide audience compared to those generated by the company. In contrast, Le et al. [22] point out that online customer relationships can be strengthened by offering relevant and valuable content and that social networks can influence various dimensions of a relationship, potentially fostering closer relationships with customers. However, as Steinhoff et al. [23] posit, many e-commerce platforms require customers to electronically submit and share personal information and payment details, which has meant that some customers face significant trust issues. As a result, it is common in the development of digital channels to consider the need to safeguard customers' private information and ensure transactional security as an aspect of great concern. It should also be considered that the information shared by companies on the Internet, related to their value proposition, can generate trust according to the signaling theory. Thus, digital channels should be oriented to the creation and maintenance of relationships based on the trust of the financial consumer because, to the extent that this is achieved, "it is more likely that customers are more committed to e-commerce and online businesses" (Thaichon et al. [24]) (p. 9).

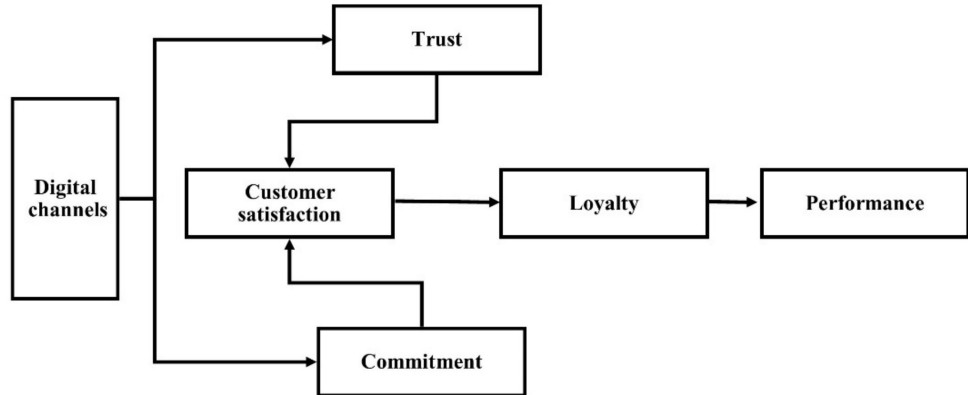

**Figure 1.** Relationship between digital channels and business performance. Note: loyalty building based on trust and commitment in financial technologies. Source: own elaboration based on Minta [20].

### 3. Methodology

This study aims to characterize the positioning and web traffic of Colombian banking establishments. To achieve this objective, the development of a study with a quantitative approach, non-experimental design, and descriptive scope is proposed [25]. Data will be collected related to the development of digital channels by banking establishments that, according to Decree No. 663 of 1993 [26], are institutions whose primary function is the collection of resources in current bank accounts and other demand or term deposits to perform active credit operations. The list of institutions supervised by the Financial Superintendency of Colombia and operating as of December 2021 is obtained from the respective web page. A characterization of the results of the digital marketing effort of the credit establishments is made based on metrics such as global and national ranking, number of unique visits, bounce rate, and the origin of the traffic to the website. In addition, the trends in revenue generation by Colombian banking establishments and the number of visits to the website will be analyzed in an exploratory manner.

According to information from the Financial Superintendence of Colombia [27], there are 28 duly regulated banking establishments in Colombia. The information that will be the subject of analysis is provided by Similarweb, which is a platform that extracts billions of digital signals every day from what is happening online, "leveraging artificial intelligence algorithms honed by a team of over 200 top data scientists and 50 PhDs" [28]. The data captured have to do with online performance metrics of the websites of financial institutions, given the significant increase of digital users, since "with the digitization of the financial sector in Colombia there is a change in the perspective of banking as a traditional financial institution and service provider, where experience and personalization move to an important plane" (Jimenez, as cited in Revista Semana [29]). The use of the tool Similarweb conditions the research since the site does not present data from websites that receive less than 5000 views per month, so the sample is reduced to 20 banking establishments. Therefore, convenience sampling is used due to the availability of information for some banking establishments on the platform under analysis. Although convenience sampling can cause some selection bias [30], the sample corresponds to 71.84% of the population for this study. To process the data, SPSS software version 20 will be used, which is characterized by its "ease of use, together with the power and integrity of the software, making it one of the most powerful tools" [31] (p. 5). We will take advantage of the benefits offered by statistical graphs and the analysis of frequency tables to interpret the patterns and trends that characterize the use of digital channels in the banking sector.

The categories considered to carry out the analysis are related to the digital channels that banking establishments leverage to create and maintain relationships with their customers. According to Le et al. [22], the predominant trends in technologies for digital marketing management include social media platforms, bilateral marketplaces, immersive

technologies, artificial intelligence, blockchain, and the Internet of things. For their part, Thaichon et al. [24] argue that many new technologies, such as mixed reality, big data, and artificial intelligence, have changed the way marketing relationships work. Meanwhile, with specific regard to the field of financial services, "technologies such as artificial intelligence, machine learning, and Blockchain introduce new possibilities for offering financial activities" [1] (p. 3). In addition, the dizzying development of information technology and the Internet has been an enhancer of the innovation of banking establishments in the digital context, making it possible to develop new channels that allow forging and managing relationships with the financial consumer. In this regard, studies in the academic literature show that, in addition to ATMs, POS, and telephone banking, "there is a significant correlation between new banking service platforms and the bank's market share" (Nazaritehrani and Mashali [16]) (p. 14).

For this study, the categories on which the analysis will be carried out are direct marketing, referrals, web traffic, social networks, email, and display advertising, whose information is provided by the Similarweb platform. Additionally, online performance metrics such as the number of visitors to the website, bounce rate, pages per visit, and average duration of each visit will be taken into account for the analysis, since it has been shown that "the technical performance of the website is positively related to the flow, which consists of the customer's engagement with the website" [7] (p. 507). The Similarweb platform has been used in different studies to analyze positionings, such as in the tourism sector [32], behavioral studies [33], government and social institutions [34], and the agrifood sector [35]. Regarding the financial sector, no studies were identified where this tool is used to analyze the positioning of the websites of financial institutions; therefore, this work generates a contribution in that sense.

## 4. Results

To characterize the use of electronic channels by Colombian banking establishments, a statistical analysis is applied, which includes the consolidation of information in frequency tables and their respective systematization with descriptive statistics and bar and sector diagrams. Additionally, a bivariate analysis is carried out, applying contingency tables and the analysis of independence through the non-parametric chi-square $\chi^2$ test. Most of the data are consolidated cross-sectional data, limited to 20 banks, and where there is no knowledge of their probabilistic distribution. For some analyses, it is necessary to weigh by frequency of visits to the website to increase the amount of data required for applying statistical techniques. Finally, a simple correspondence analysis is performed, a multivariate statistical analysis technique that allows the dimensional reduction of the classification of objects on a set of attributes and their representation in a perceptual map [36]. Correspondence analysis is expected to complement the analysis of relationships between variables so that the underlying dimensions or attributes can also be identified to characterize these relationships. According to Abdi and Béra [37], correspondence analysis transforms a data table into two sets of factor influences, one for the rows and one for the columns. Furthermore, capabilities can be drawn as maps that represent essential information. According to Greenacre [38], correspondence analysis has increased in popularity in different fields, contributing to the development of studies in areas such as sociology, ecology, paleontology, archaeology, geology, education, medicine, biochemistry, linguistics, market research, advertising, religious studies, philosophy, art, and music.

### 4.1. General Characterization of Website Traffic

Generally, the positioning of web pages is measured through rankings or classifications provided by benchmark providers, such as Alexa or Similarweb. These sites use metrics of quality or quantity of users to rank different sites. For example, Similarweb [39] performs the ranking by combining estimates of monthly unique visitors to a site and monthly page views on mobile and desktop traffic. As it is, when the aggregate of these two elements is higher, the ranking that this platform achieves on the website will also be higher. According

to Hung [40], SEO is a process of creating an identity in the minds of Internet users, and a website functions as a showcase that gives users the first impression and then influences their perception. For his part, Wang [41] mentions that some studies analyze the attributes that affect service quality, online store image, and brand image, mentioning aspects such as ease of use, website design, financial security, privacy, and trust, as well as interactivity, customization, and personalization. Considering the above, a characterization of the web positioning of Colombian banking establishments is presented below.

The first analysis carried out is related to the distribution by gender of the traffic arriving at the website of Colombian banking establishments. In general, access by female users is more prevalent than access by male users. These differences are particularly noticeable in access to the websites of Banco Agrario de Colombia (56.20%), Banco Pichincha (55.87%), Ser Finanzas (55.41%), and Banco Mundo Mujer (55.08%), where there is more traffic to the website by female users. In contrast, in the case of Lulo Bank, most of the traffic is male (53.54%). As shown in Table 1, applying the chi-square contingency statistic ($\chi^2_{19} > 10$) with nineteen degrees of freedom, it is evident that there is a statistically significant relationship between the banking establishment variable and the gender of the users from which the web traffic originates.

**Table 1.** Chi-square test gender and banking establishment.

|  | Value | df | Asymp. Sig. (2-Sided) |
|---|---|---|---|
| Pearson Chi-Square | 25.073,921 [1] | 19 | 0.000 |
| Likelihood Ratio | 25.078,745 | 19 | 0.000 |
| Linear-by-Linear Association | 671.403 | 1 | 0.000 |
| No Valid Cases | 69.394.654 |  |  |

[1] Zero cells (0.0%) have an expected frequency of less than 5.

Figure 2 shows the number of users by age group for the twenty banks under analysis, for Banco de Bogotá, which was the first financial institution created in the country in 1870, and Lulo Bank, which the Financial Superintendency of Colombia endorsed to start operations in 2021 as a 100% digital bank. In principle, it is observed that the most significant amount of traffic to the websites of banking establishments comes from young users in the age range of 18 to 34 years. According to Nejad [1], unlike traditional financial institutions, emerging and leveraged fintech ones do not have a large customer base, but they present convenience at low costs, and this flexibility, convenience, and low-cost appeal to younger generations. However, the results seem counterintuitive in the sense that no relevant differences are perceived between Banco de Bogotá, which could be considered as the traditional banking establishment in the country, and Lulo Bank, a bank with an innovative business model and high flexibility in terms of transaction costs.

*4.2. Characterization of Website Positioning*

According to what is shown in Figure 3, most of the web portals of Colombian banking establishments are located among the best-positioned sites in the world according to the ranking of Similarweb for the month of March, that is, the time when the consultation was carried out. This result is remarkable because, even if a query is made for the Financial Industry category, it will be found that a website such as bancolombia.com (accessed on 17 april 2022) is among the first one hundred sites related to that specific area, considering that it is ranked number 2451 worldwide.

Considering what is observed in Figure 4, most Colombian banking establishments rank from position 22 to position 1549 of the best-positioned pages in the country. Among the fifty best-positioned websites in Colombia, at least two stand out as corresponding to Colombian banking establishments: bancolombia.com and davivienda.com. This list is led by international websites such as the search engine google.com, the social network for multimedia content YouTube, and the site of the instant messaging application WhatsApp.

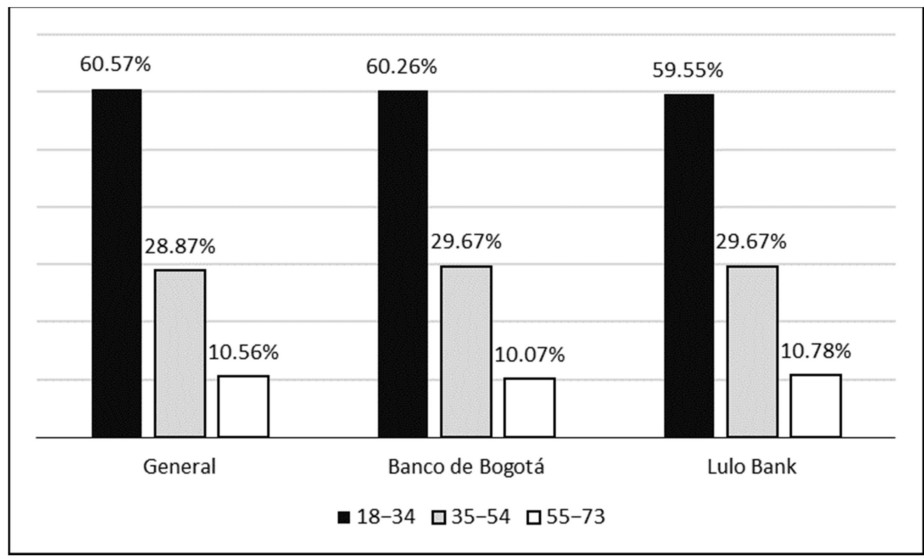

**Figure 2.** Traffic to websites of banking establishments, according to age. Note: Similarweb classifies information into six age groups. For this study, these groups are reduced to three (3) categories only to facilitate the presentation of the information and interpretation.

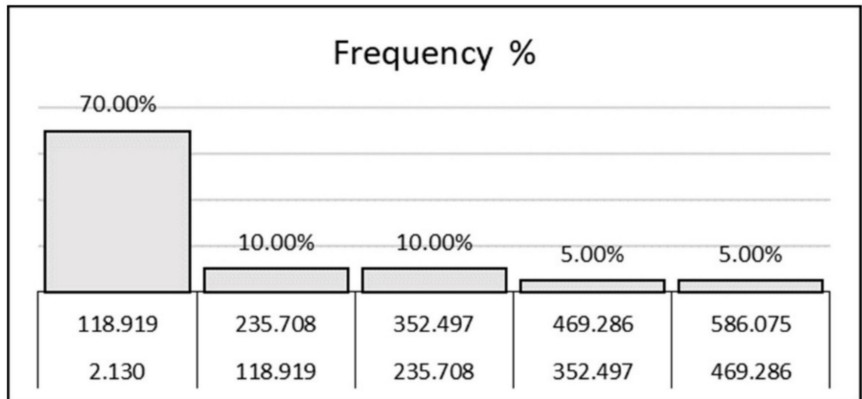

**Figure 3.** Worldwide positioning ranking. Note: five (5) groups are formed, and the frequency is shown in the figure.

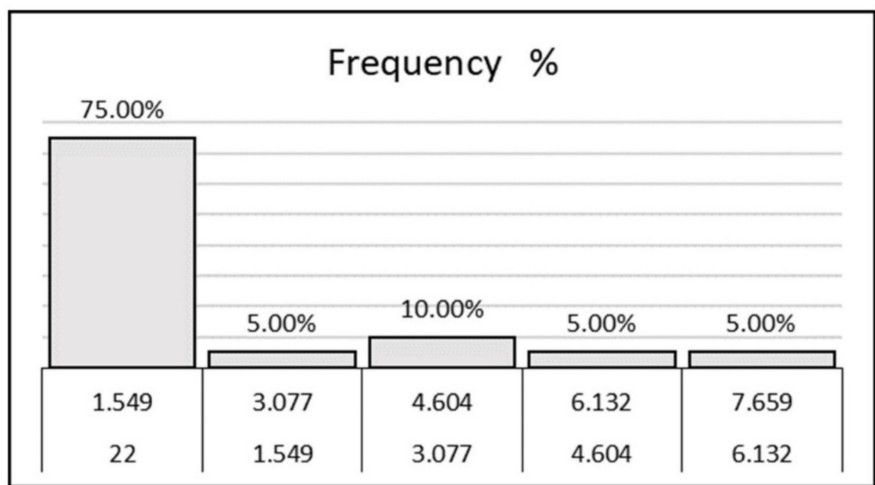

**Figure 4.** National positioning rankings. Note: five (5) groups are formed, and the relative frequency is shown in the figure.

As shown in Figure 5, at least fifty percent of Colombian banking establishments manage a bounce rate lower than 42.35%. According to Singal and Kohli [33], this rate is measured as the number of users who abandon the website within ten seconds of arrival. Regarding the measurement made by Similiarweb [28], it is the average percentage of visitors who view a single page before leaving the website. Thus, in this platform, the bounce rate is related to single-page visits, which is calculated by "dividing the single page visits by the total visits to your site during a given period" [39]. The bounce rate is reported as a percentage of total visits during the specified time. There is no theoretically defined criterion for what could be considered a high bounce rate, but by thumb rule, it could be defined as above 50% [42]. As it is, half of the banking establishments have acceptable bounce rates.

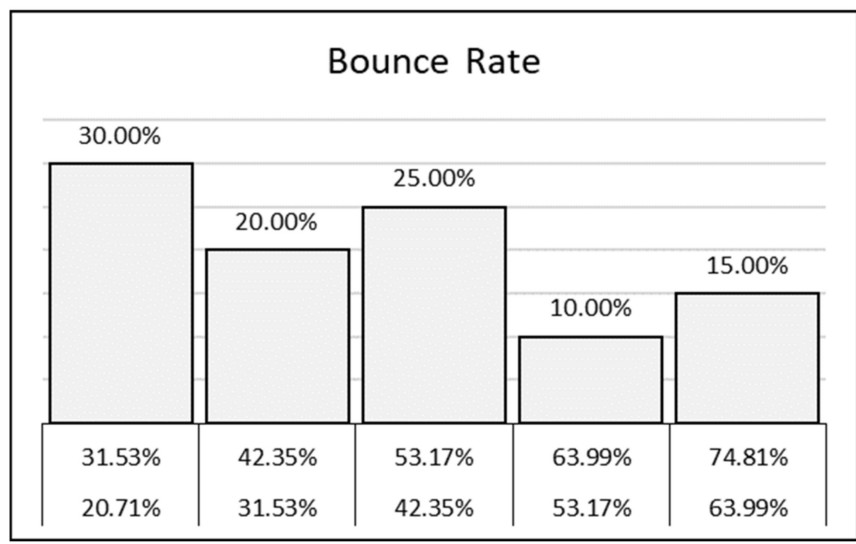

**Figure 5.** Website bounce rate. Note: five (5) groups were formed, and the relative frequency is shown in the figure.

Table 2 shows positive correlations between the revenue growth of banking establishments with duration and bounce rate. Although it is not possible to obtain statistically significant evidence of the relationship between revenue growth and the overall classification (positioning ranking) of the banking establishment (based on data from the income from operations accounting account extracted from the Financial Statements under IFRS available on the web page of the Financial Superintendence of Colombia), a positive correlation of 19.8% is obtained, which indicates, at a descriptive level, that the better-positioned establishments obtained greater growth in their operating results during the month of December 2021. On the other hand, with the rebound rate, there is evidence of a statistically significant correlation at a 90% confidence level ($t_C > 1.73$), which is positive and corresponds to 38.6%. Similarly, a positive and significant correlation at a 90% confidence level ($t_C > 1.73$) is observed between the ranking and the rebound rate.

As shown in Figure 6, although a certain relationship is visible in how the trend in the number of unique visitors to the site and the banking industry (consisting of the twenty banks on which the data were extracted, out of a total of twenty-eight) revenues evolves, there is not a full correspondence. For example, while the number of visitors to websites decreased in December, the level of revenues from banking establishments increased compared to the immediately preceding period. This may indicate that many of the sources of revenue for Colombian banking establishments come from services provided through means other than the main web platform.

**Table 2.** Correlations between revenue growth in December 2021 with bounce rate and average visitor duration.

| | | Revenue_Growth | Ranking | Bounce_Rate |
|---|---|---|---|---|
| Revenue_Growth | Pearson Correlation | 1 | 0.198 | 0.394 |
| | Sig. (2-tailed) | | 0.402 | 0.085 * |
| | N | 20 | 20 | 20 |
| Ranking | Pearson Correlation | 0.198 | 1 | 0.386 |
| | Sig. (2-tailed) | 0.402 | | 0.093 * |
| | N | 20 | 20 | 20 |
| Bounce_Rate | Pearson Correlation | 0.394 | 0.386 | 1 |
| | Sig. (2-tailed) | 0.085 * | 0.093 * | |
| | N | 20 | 20 | 20 |

* Correlations are significant at the 10% level (2-tailed). Source: data obtained from Superintendencia Financiera de Colombia December 2021, Similarweb, and systematized with SPSS v20.

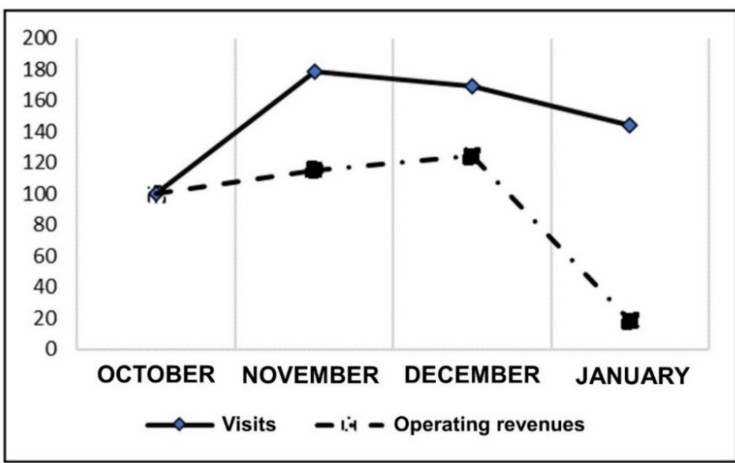

**Figure 6.** The trend in website visits and operating revenues. Note: data were normalized on a base of 100, where the initial value is the number of unique visitors and operating revenues for October 2021.

### 4.3. Characterization of Traffic Sources to the Website

According to Table 3, most of the traffic to the websites of Colombian banking establishments is direct or through search engines. In the case of direct traffic, Similiarweb [28] indicates that it has to do with the traffic of users who enter directly to the URL, use a bookmark or a saved link, etc., which can be used as a barometer of the strength of the brand as it provides information on awareness and demand. Meanwhile, traffic through search engines has to do with access to the website using keywords and cannot be influenced by paid advertising, so "when there is a correlation with direct traffic, it indicates strong brand awareness, as many organic visits are generated by brand terms" [43]. According to the above and considering that direct traffic and traffic through search engines are prevalent, it can be inferred that the positioning of the websites of banking establishments is also related to the positioning of their brand.

Table 4 shows the traffic from social networks to the websites of Colombian banking establishments. Although traffic from these platforms represents less than 1% of total traffic, the importance of social networks in promoting interactions between the brand and the users and among the users themselves around the brand should not be underestimated. As Singh, Chakraborty, Bose Biswas, and Majumdar [2] argue, social networks have changed the whole range of business, becoming universal and ubiquitous, and they are becoming a powerful means of communication between banks and their customers. For this case, the prevalence is in the instant messaging platform WhatsApp, followed by YouTube, Facebook, and Twitter. According to Similarweb [43], a website that generates consistently high traffic from social networks is likely to have a loyal user community.

**Table 3.** Website traffic origin.

| Channel | Absolute Frequency | Relative Frequency |
|---|---|---|
| Direct Traffic | 32,902,184 | 47.41% |
| Referrals | 5,527,362 | 7.97% |
| Search Engine | 29,812,657 | 42.96% |
| Social Media Traffic | 382,345 | 0.55% |
| Email Traffic | 656,492 | 0.95% |
| Display Advertising Traffic | 113,614 | 0.16% |
| Total | 69,394,655 | 100.00% |

Note: data are consolidated traffic source data extracted from Similarweb.

**Table 4.** Traffic obtained through social networks.

| Channel | Absolute Frequency | Relative Frequency |
|---|---|---|
| WhatsApp | 166,999 | 43.68% |
| YouTube | 125,559 | 32.84% |
| Facebook | 60,180 | 15.74% |
| Twitter | 11,098 | 2.90% |
| LinkedIn | 11,934 | 3.12% |
| Instagram | 3,322 | 0.87% |
| Meetup | 719 | 0.19% |
| Other | 2,534 | 0.66% |
| Total | 382,345 | 100.00% |

Note: data are consolidated traffic source data extracted from Similarweb and focused on social networks.

Figure 7 shows the perceptual map derived from a simple correspondence analysis, which relates the classification of banks according to their online positioning in the categories low (<300), medium (between 300 and 1300), and high (>1300) and the origin of traffic to their websites. According to Table A1 (Appendix A), the dimension that contributes significantly to the characterization of credit institutions is the first, with a contribution of 96.4%. Therefore, the analysis will focus on this dimension, which has been assigned the positioning label. Considering the factors related to this dimension, two groups can be distinguished. On the one hand, there is a group with an important level of direct traffic to the website, referrals, and email traffic. These are banking establishments with a medium level of positioning; therefore, they can be identified as those that have achieved an important level of positioning; therefore, users access their website directly through other portals that have a direct link to the bank's page or even through email. On the other hand, there are emerging banks, i.e., those that have not achieved the same level of positioning, so that traffic to their websites comes from other sources such as search engines, social networks, or paid advertising. Therefore, these unpositioned banks must make efforts in terms of digital advertising campaigns, in improving their organic positioning or in the management of their social networks to captivate the attention of users to access their portals. It should be noted that, although a statistically significant correlation is obtained between the size of Colombian banking establishments measured by asset level as of December 2021, with respect to the positioning ranking, this significance is only for a ($p < 10\%$). The correlation is −39.01%, so it is expected that the more activity a bank has, the more privileged position in the positioning ranking (first positions, less is better) will be occupied.

Figure 8 shows the perceptual map that relates the classification of banks according to their web positioning and the origin of traffic to their websites. As shown in Table A3 (Appendix A), the dimension that contributes significantly to the characterization of credit institutions is the first, with a contribution of 78.1%. However, in this case, the second dimension also has a contribution that can be considered for the analysis. These dimensions are assigned the labels of diversification and traditional. Thus, two groups can be distinguished. In the first group are those credit establishments with a low positioning and that use specific social networks, generally Facebook. On the other hand, in the group of

banking establishments with high positioning are those that have diversified their activity on social networks, which allows them to obtain traffic from networks such as Twitter, LinkedIn, or WhatsApp. The second dimension separates high-positioning banking establishments from those of medium positioning. The former makes extensive use of traditional microblogging networks such as Twitter or those that make it possible to share multimedia content, such as YouTube, among others. Meanwhile, some banking establishments are using other types of social networks, such as Instagram or Meetup, where they can share graphic content and advertise at a lower cost or even become involved in communities with specific interests and whose users are willing to transfer their interactions to real scenarios.

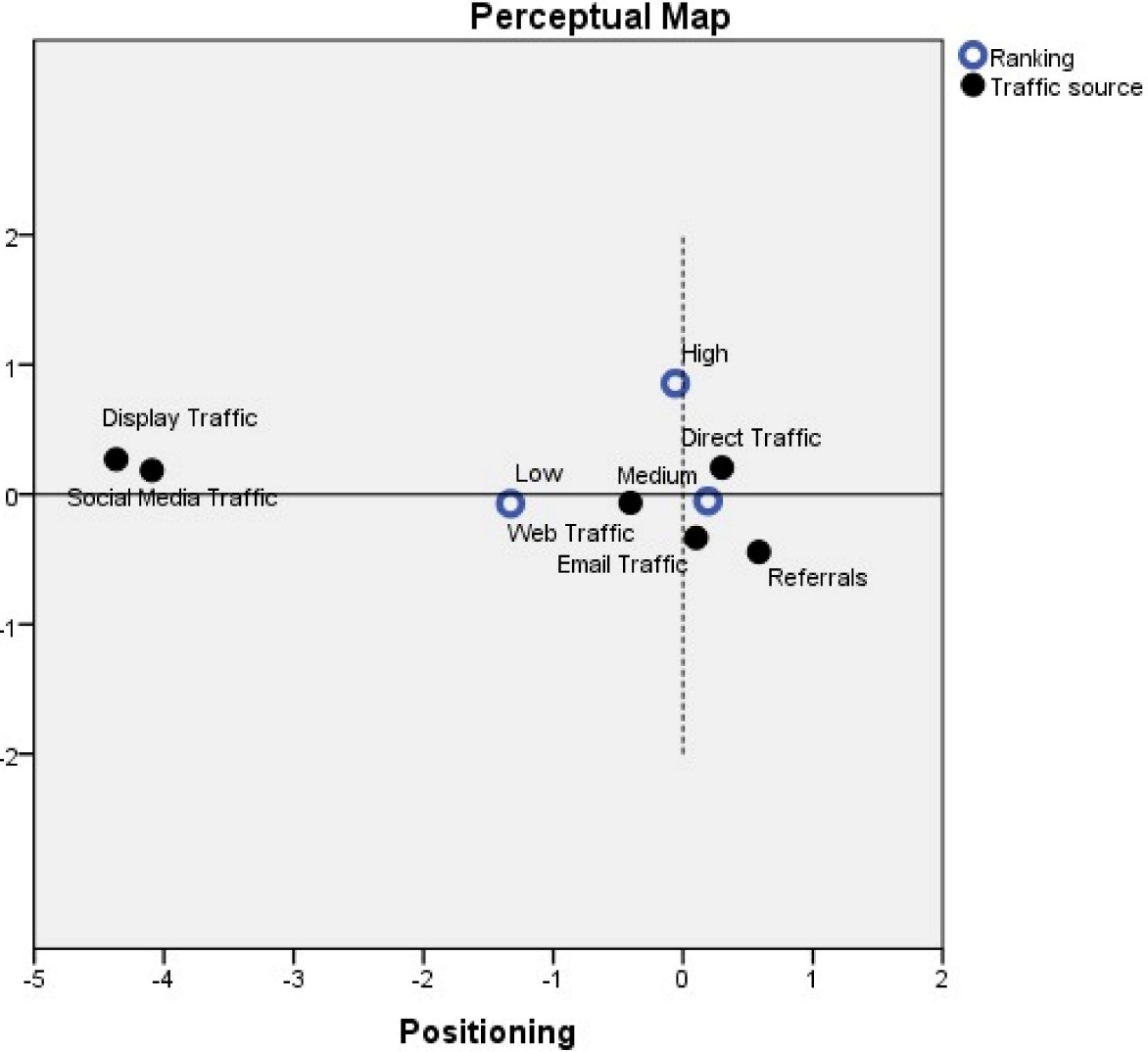

**Figure 7.** Correspondence analysis between positioning and web traffic source. Note: constructed from consolidated and weighted data from the Similarweb platform.

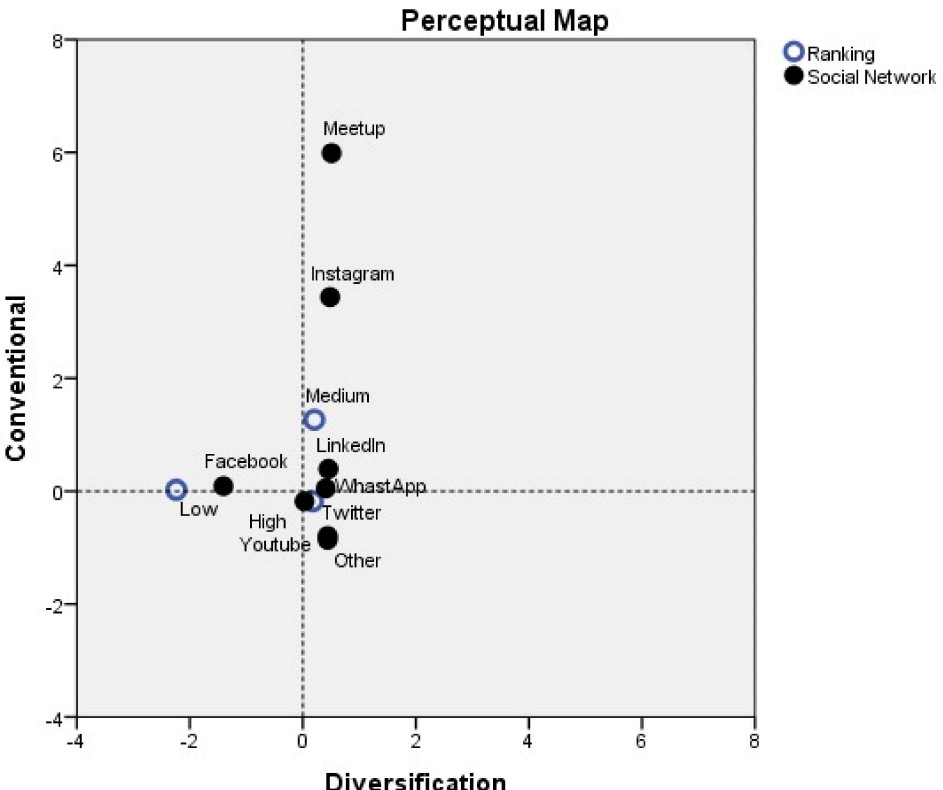

**Figure 8.** Correspondence analysis between positioning and traffic to the portal obtained from social networks. Note: constructed from consolidated and weighted data from the Similarweb platform.

## 5. Discussion

Currently, digital strategy is a key issue for competitiveness in different industries, including finance. According to Amelda et al. [44], studies show that digital marketing allows the banking sector to reach unreached customers or those who do not have the banking services that fit their needs; in addition, customers also want to see content online which matches the solutions they are looking for in these institutions. The main Colombian banks have a robust physical infrastructure that significantly leverages the operation in such a way that branches, and ATMs are a factor to evaluate when identifying the banking establishment in which a product will be opened. According to Gómez Villamizar [45], according to a Mambu study of 1250 Latin American millennials and Gen Z, in Colombia, 79% of those surveyed opted for traditional banking, although loyalty is not guaranteed, as 42% then indicated that they would be willing to change if they were offered better conditions and benefits. Thus, the opportunity is given for the Colombian financial industry to diversify, and proof of this is the entry of the famous neo banks such as Lulo Bank or NuBank. With the foregoing, although today digital solutions are just a factor that complements the strategy of banking establishments, it is expected that in the future, it will become a key factor of competitiveness, in accordance with advances in the features, versatility, and security of digital platforms.

This study was carried out through a descriptive characterization process on the positioning of the websites of Colombian banking establishments and the sources of traffic to these sites. First, a higher prevalence of female audiences is observed in the access to the sites of the banking establishments, and relevant differences are found on some pages with respect to others. This shows that brand positioning is a crucial aspect to obtain deliberately planned web traffic, such as, for example, Banco Mundo Mujer, which promotes financial inclusion and the empowerment of women with its range of products and services. However, it is not the financial institution with the highest female web traffic, which leads us to think that other banking establishments have generated a value offer

that is also attractive to Internet users of this gender. On the other hand, there is a higher prevalence of users between 18 and 34 years of age; therefore, it is very important that the websites of the banking establishments tend to manage an interface that favors the attention and enjoyment of a good experience for younger users. A result that is a source of disappointment is that there were no significant differences between the web traffic of traditional banking establishments such as Banco de Bogotá and the new digital banking, whose benchmark in Colombia is Lulo Bank. This is a challenge for innovative business models in the financial industry since what characterizes these emerging banks leveraged on financial technology is that they do not have a large customer base, but they offer convenience at low costs, and this flexibility, convenience, and low cost attract the younger generations [1].

When analyzing the positioning of the websites of Colombian banking establishments, there is a significant dispersion in the classification assigned to them by Similarweb with its ranking, but, in general, most of the banking establishments occupy privileged places in the ranking of web traffic. The above shows the results of the efforts made by banking establishments in the digital area to achieve greater public traction and a better experience on their website. This is important, as positioning creates an identity for Internet users, and the website functions as a showcase that provides the first impression and influences their perception [40]. Additionally, many of the websites of banking establishments have acceptable bounce rates (below 50%), which is an indicator of the quality of the website. This is a user retention metric, which is expected to be low; otherwise, it would indicate that people are abandoning a website and are not willing to stay and explore [39]. Thirty percent of Colombian banking establishments have a bounce rate of less than 32%, which is quite favorable compared to the average bounce rate of the fifty best-positioned websites worldwide, around 34%.

Finally, this study characterizes the sources of web traffic for Colombian banking establishments. Using a multivariate analysis approach and with the support of a perceptual map, differences can be seen between medium and low-positioned banking establishments in terms of traffic source. In the case of banking establishments with a medium positioning, located between 3.00 and 1.300 in the ranking of Similarweb for Colombia, most of the web traffic comes from search engines and referrals. This is a sign of good results in terms of organic positioning or brand awareness and a good affiliate marketing strategy or the enjoyment of significant media coverage [43]. Meanwhile, something that differentiates the web traffic of banking establishments with lower web positioning (ranking above 1300) is that part of it comes from paid digital advertising or through social networks, which leads to inferences about the efforts that banking establishments carry out with the aim of achieving brand recognition. Subsequently, attention is focused on the web traffic coming from social networks, although it is quite inferior compared to other channels, and in the case of banking establishments, it cannot be ignored that they are media that are changing the whole range of business, becoming increasingly universal and ubiquitous and a source of communication between banks with their customers [2]. In the end, it is observed that most of the web traffic from social networks comes from traditional networks such as Facebook, WhatsApp, and YouTube; although well, some establishments have diversified to a better extent than others the sources of traffic to their website and, even for some, traffic comes from less recurrent platforms such as Meetup.

In Colombia, significant efforts are being made in regulatory matters to strengthen the financial industry by developing and applying new technologies. Proof of this is the implementation of a controlled test space (ECP), which is a public innovation tool that seeks to strengthen the capacities of the state to adjust the regulatory framework to the new market dynamics, helping to promote safe and sustained financial innovation [27]. Thus, it is expected that research related to the web positioning of financial institutions will be developed in greater depth as increasingly sophisticated technologies are being adopted, such as big data, machine learning, natural language processing, cloud computing, blockchain technology, and chatbots, among others, which are changing the way advice is

given, service is provided, and financial transactions are carried out. For the above, the study developed has limitations, which are expected to be used to develop further studies; for example, the possibility of expanding the sample size, working with longitudinal data instead of cross-sectional, as well as transcending geographical boundaries and industry level, since this study was concentrated in Colombia and only in banking establishments. In terms of practical implications, this work relates some web presence and positioning metrics, which can be used to monitor the digital strategy of financial institutions and other areas of economic activity. This analysis can be given context by taking advantage of the benefits of technological platforms such as Similarweb or Alexa, with which comparisons can be made with other websites inside and outside the industry.

## 6. Conclusions

The findings of this study elucidate that a good part of the websites of banking establishments are well positioned in comparison with others, in addition to handling low bounce rates. In addition, it is possible that a significant portion of this traffic comes from individuals between 18 and 34 years of age and the female gender. Likewise, web traffic comes primarily from direct access to the portal or search engines. Although social networks have a marginal representation, it was possible to show that traffic from these platforms comes from traditional networks such as Facebook, WhatsApp, and Twitter. In the end, better profiling of users expected to reach the website is recommended, considering demographic and cultural characteristics, as well as deepening efforts in organic positioning and social networks to attract the attention of the public that best fits the characteristics of the brand. However, future studies should analyze whether the actions aimed at strengthening the digital strategy can significantly affect the cost structure and evaluate whether marginal revenues justify the effort to achieve a higher level of positioning.

A slight but not negligible correlation is found between web positioning and the size of a bank, measured by the level of assets. Therefore, today, experience, infrastructure, and even network effects favor banking competitiveness in the digital context. The results show that, in the case of establishments with lower online positioning, additional efforts are required in the field of digital marketing, so they should seek this positioning by improving organic traffic, which is obtained through social media or paid advertising. Meanwhile, the portals of the best-positioned banking establishments enjoy direct traffic or websites that refer to the portal. However, new business models with an eminently digital approach are currently entering the Colombian financial industry, as is the case of Lulo Bank, which recently received the green light from the Colombian Financial Superintendence to operate as a financing company [46]. This new trend, known as the neo banks, will significantly dispense with the traditional infrastructure of bank branches.

Finally, the results indicate that social networks generate a marginal contribution to the web traffic that reaches the portals of Colombian banking establishments. Despite the above, social networks increasingly generate greater added value in the digital strategy. Therefore, it is a finding that invites reflection, considering the dynamics of Fintech, since in works such as Sakas et al. [47] it is observed that a gradual increase in analytical metrics of social networks leads to better website traffic and lower organic campaign costs. Considering the trend toward digitization of banking services, it is convenient to evaluate how the benefits of these networks can be better exploited. Thus, in later studies, the unit of analysis could be extended to include financial institutions, regardless of their category <beyond banking establishments>, to study the repercussions that social networks can have on the web traffic of these businesses.

**Author Contributions:** Conceptualization, J.S.R.R., A.R.R.T., A.M.M.M. and J.C.A.-P.; methodology, J.S.R.R., A.R.R.T., A.M.M.M. and J.C.A.-P.; software, J.S.R.R., A.R.R.T., A.M.M.M. and J.C.A.-P.; validation, J.S.R.R., A.R.R.T., A.M.M.M. and J.C.A.-P.; formal analysis, J.S.R.R., A.R.R.T., A.M.M.M. and J.C.A.-P.; investigation, J.S.R.R., A.R.R.T., A.M.M.M. and J.C.A.-P.; resources, J.S.R.R., A.R.R.T., A.M.M.M. and J.C.A.-P.; data curation, J.S.R.R., A.R.R.T., A.M.M.M. and J.C.A.-P.; writing—original draft preparation, J.S.R.R., A.R.R.T., A.M.M.M. and J.C.A.-P.; writing—review and editing, J.S.R.R., A.R.R.T., A.M.M.M. and J.C.A.-P.; visualization, J.S.R.R., A.R.R.T., A.M.M.M. and J.C.A.-P. All authors have read and agreed to the published version of the manuscript.

**Funding:** This research received no external funding.

**Institutional Review Board Statement:** Not applicable.

**Informed Consent Statement:** Not applicable.

**Data Availability Statement:** The financial data used in this study is available at Superintendencia Financiera de Colombia https://www.superfinanciera.gov.co/. Data on financial metrics were obtained from: https://www.similarweb.com/es/.

**Acknowledgments:** The authors thank the Universidad Nacional Abierta y a Distancia—UNAD.

**Conflicts of Interest:** The authors declare no conflict of interest.

## Appendix A

**Table A1.** Summary of correspondence. Positioning and source of web traffic.

| Dimension | Singular Value | Inertia | Chi Square | Sig. | Proportion of Inertia | | Confidence Singular Value | |
|---|---|---|---|---|---|---|---|---|
| | | | | | Accounted for | Cumulative | Standard Deviation | Correlation 2 |
| 1 | 0.236 | 0.056 | | | 0.964 | 0.964 | 0.000 | −0.020 |
| 2 | 0.045 | 0.002 | | | 0.036 | 1.000 | 0.000 | |
| Total | | 0.058 | 59,157,798.904 | 0.000 * | 1.000 | 1.000 | | |

\* Ten degrees of freedom.

**Table A2.** Overview of correspondence score by row. Positioning and source of web traffic.

| Ranking | Mass | Score in Dimension | | Inertia | Contribution | | | | |
|---|---|---|---|---|---|---|---|---|---|
| | | 1 | 2 | | Of Point to Inertia of Dimension | | Of Dimension to Inertia of Point | | |
| | | | | | 1 | 2 | 1 | 2 | Total |
| High | 0.058 | −0.059 | 0.856 | 0.002 | 0.001 | 0.941 | 0.024 | 0.976 | 1.000 |
| Medium | 0.825 | 0.192 | −0.051 | 0.007 | 0.128 | 0.046 | 0.987 | 0.013 | 1.000 |
| Low | 0.116 | −1.330 | −0.071 | 0.049 | 0.871 | 0.013 | 0.999 | 0.001 | 1.000 |
| Active Total | 1.000 | | | 0.058 | 1.000 | 1.000 | | | |

**Table A3.** Overview of correspondence score by column. Positioning and web traffic source.

| Source | Mass | Score in Dimension | | Inertia | Contribution | | | | |
|---|---|---|---|---|---|---|---|---|---|
| | | 1 | 2 | | Of Point to Inertia of Dimension | | Of Dimension to Inertia of Point | | |
| | | | | | 1 | 2 | 1 | 2 | Total |
| Direct Traffic | 0.418 | 0.299 | 0.206 | 0.010 | 0.159 | 0.390 | 0.917 | 0.083 | 1.000 |
| Referrals | 0.123 | 0.584 | −0.443 | 0.011 | 0.178 | 0.533 | 0.900 | 0.100 | 1.000 |
| Web Traffic | 0.442 | −0.405 | −0.064 | 0.017 | 0.307 | 0.040 | 0.995 | 0.005 | 1.000 |
| Social Network Traffic | 0.003 | −4.368 | 0.270 | 0.016 | 0.282 | 0.006 | 0.999 | 0.001 | 1.000 |
| Email Traffic | 0.013 | 0.100 | −0.335 | 0.000 | 0.001 | 0.031 | 0.317 | 0.683 | 1.000 |
| Traffic Display Advertising | 0.001 | −4.092 | 0.186 | 0.004 | 0.074 | 0.001 | 1.000 | 0.000 | 1.000 |
| Active Total | 1.000 | | | 0.058 | 1.000 | 1.000 | | | |

**Table A4.** Summary of correspondence. Positioning and traffic obtained from social networks.

| Dimension | Singular Value | Inertia | Chi Square | Sig. | Proportion of Inertia | | Confidence Singular Value | |
|---|---|---|---|---|---|---|---|---|
| | | | | | Accounted for | Cumulative | Standard Deviation | Correlation |
| | | | | | | | | 2 |
| 1 | 0.399 | 0.159 | | | 0.781 | 0.781 | 0.002 | 0.018 |
| 2 | 0.211 | 0.045 | | | 0.219 | 1.000 | 0.002 | |
| Total | | 0.204 | 77,999.898 | 0.000 * | 1.000 | 1.000 | | |

\* 10 degrees of freedom.

**Table A5.** Overview of correspondence score by row. Positioning and traffic obtained from social networks.

| Ranking | Mass | Score in Dimension | | Inertia | Contribution | | | | |
|---|---|---|---|---|---|---|---|---|---|
| | | 1 | 2 | | Of Point to Inertia of Dimension | | Of Dimension to Inertia of Point | | |
| | | | | | 1 | 2 | 1 | 2 | Total |
| High | 0.811 | 0.175 | −0.182 | 0.016 | 0.062 | 0.127 | 0.636 | 0.364 | 1.000 |
| Medium | 0.115 | 0.203 | 1.264 | 0.041 | 0.012 | 0.873 | 0.046 | 0.954 | 1.000 |
| Low | 0.074 | −2.240 | 0.023 | 0.148 | 0.926 | 0.000 | 1.000 | 0.000 | 1.000 |
| Active Total | 1.000 | | | 0.204 | 1.000 | 1.000 | | | |

**Table A6.** Overview of correspondence score by column. Positioning and traffic obtained from social networks.

| Red_Social | Mass | Score in Dimension | | Inertia | Contribution | | | | |
|---|---|---|---|---|---|---|---|---|---|
| | | 1 | 2 | | Of Point to Inertia of Dimension | | Of Dimension to Inertia of Point | | |
| | | | | | 1 | 2 | 1 | 2 | Total |
| Facebook | 0.157 | −1.407 | 0.089 | 0.125 | 0.780 | 0.006 | 0.998 | 0.002 | 1.000 |
| Instagram | 0.009 | 0.481 | 3.437 | 0.022 | 0.005 | 0.486 | 0.036 | 0.964 | 1.000 |
| LinkedIn | 0.031 | 0.450 | 0.392 | 0.004 | 0.016 | 0.023 | 0.714 | 0.286 | 1.000 |
| Meetup | 0.002 | 0.507 | 5.987 | 0.014 | 0.001 | 0.319 | 0.013 | 0.987 | 1.000 |
| Other | 0.007 | 0.438 | −0.861 | 0.002 | 0.003 | 0.023 | 0.328 | 0.672 | 1.000 |
| Twitter | 0.029 | 0.438 | −0.794 | 0.006 | 0.014 | 0.087 | 0.366 | 0.634 | 1.000 |
| WhatsApp | 0.437 | 0.405 | 0.048 | 0.029 | 0.180 | 0.005 | 0.993 | 0.007 | 1.000 |
| YouTube | 0.328 | 0.029 | −0.182 | 0.002 | 0.001 | 0.051 | 0.046 | 0.954 | 1.000 |
| Active Total | 1.000 | | | 0.204 | 1.000 | 1.000 | | | |

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
