# Peer review of "Positioning and Web Traffic of Colombian Banking Establishments"

_jtaer, doi:10.3390/jtaer17040074_

Round 1

Reviewer 1 Report

The abstract is somewhat unconcise. Banks and the digital industry should be referred to in the abstract.

The Literature review is too extensive. This section should be shortened. It does not provide valuable scientific significance.

The methodology is unclear and the whole methodology section is vague. Moreover, the sampling is unclear (lines 294-296 “the development is not proposed”).

The conclusion is too short. More detailed results (in at least 2-3 paragraphs) must be added.

All in all, the scientific contribution of this study is insignificant. In my opinion, you should try submitting this paper in another Journal. The JTAER journal is too reputable for such research.

Nothing personal, just my professional opinion.

Reviewer 2 Report

ORIGINALITY: (*). 
- Does the paper clearly point out differences from related research? Yes
- Are the problems or approaches new? yes
For example, does the paper: address a new problem or one that has not been studied in much depth? yes
introduce an interesting research paradigm? yes
introduce an area that appears promising, or might stimulate others to develop promising alternatives? yes

SIGNIFICANCE (*).
- Is the work important? yes
- Does it advance the state of the art? yes
- Does the paper stimulate discussion of important issues or alternative points of view? yes
TECHNICAL QUALITY (*). 
- Is the paper technically sound, with compelling arguments? 
- Is there a careful evaluation? Does the paper carefully evaluate the strengths and limitations of its contributions? yes
- Does the paper offer a new form of evidence in support of or against a well known technique? 
- If the paper describes an application, is there: a clear and compelling motivation for why the chosen approach is important? a careful description of the design and implementation of the system? a thorough evaluation of the system with respect to a clearly-stated set of functional and quality requirements? yes

QUALITY OF PRESENTATION (*). 
- Is the paper clearly written? yes 
- Does the paper motivate the research? yes
- Are results clearly described and evaluated? yes
- Is the paper well organized? yes

Reviewer 3 Report

The paper addresses a current and important topic with new and interesting results.  There are only a few minor suggestions.

1. One should distinguish if the results are different for big and small banks.

2. One should discuss web traffic and non-web traffic are complements or substitutes.

3. The information provided in Table 2 should be discussed in the text.

4. Does the technology and other resources used to support web traffic increase revenue more than cost?

5. One should discuss some of the regulatory and policy implications of the results.

Round 2

Reviewer 1 Report

Thank you for addressing all the issues.